# Agents of Forest Disturbance in the Argentine Dry Chaco

Teresa De Marzo [1,2,*], Nestor Ignacio Gasparri [3], Eric F. Lambin [2,4,5] and Tobias Kuemmerle [1]

1   Geography Department, Humboldt-Universität zu Berlin, Unter den Linden 6, 10099 Berlin, Germany; tobias.kuemmerle@hu-berlin.de
2   Georges Lemaître Centre for Earth and Climate Research, Earth and Life Institute, Université Catholique de Louvain, 3, Place Louis Pasteur, 1348 Louvain-la-Neuve, Belgium; eric.lambin@uclouvain.be
3   Consejo Nacional de Investigaciones Científicas y Técnicas (CONICET), Instituto de Ecología Regional (IER), Universidad Nacional de Tucumán, CC:34, Yerba Buena CP 4107, Tucumán, Argentina; ignacio.gasparri@gmail.com
4   School of Earth, Energy & Environmental Sciences, Stanford University, 473 Via Ortega, Stanford, CA 94305, USA
5   Woods Institute for the Environment, Stanford University, 473 Via Ortega, Stanford, CA 94305, USA
*   Correspondence: teresa.demarzo@hu-berlin.de

**Abstract:** Forest degradation in the tropics is a widespread, yet poorly understood phenomenon. This is particularly true for tropical and subtropical dry forests, where a variety of disturbances, both natural and anthropogenic, affect forest canopies. Addressing forest degradation thus requires a spatially-explicit understanding of the causes of disturbances. Here, we apply an approach for attributing agents of forest disturbance across large areas of tropical dry forests, based on the Landsat image time series. Focusing on the 489,000 km$^2$ Argentine Dry Chaco, we derived metrics on the spectral characteristics and shape of disturbance patches. We then used these metrics in a random forests classification framework to estimate the area of logging, fire, partial clearing, riparian changes and drought. Our results highlight that partial clearing was the most widespread type of forest disturbance from 1990–to 2017, extending over 5520 km$^2$ ($\pm$407 km$^2$), followed by fire (4562 $\pm$ 388 km$^2$) and logging (3891 $\pm$ 341 km$^2$). Our analyses also reveal marked trends over time, with partial clearing generally becoming more prevalent, whereas fires declined. Comparing the spatial patterns of different disturbance types against accessibility indicators showed that fire and logging prevalence was higher closer to fields, while smallholder homesteads were associated with less burning. Roads were, surprisingly, not associated with clear trends in disturbance prevalence. To our knowledge, this is the first attribution of disturbance agents in tropical dry forests based on satellite-based indicators. While our study reveals remaining uncertainties in this attribution process, our framework has considerable potential for monitoring tropical dry forest disturbances at scale. Tropical dry forests in South America, Africa and Southeast Asia are some of the fastest disappearing ecosystems on the planet, and more robust monitoring of forest degradation in these regions is urgently needed.

**Keywords:** disturbance agents; disturbance regimes; forest degradation; Landsat time series; land use; LandTrendr; tropical dry forests

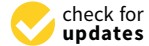



## 1. Introduction

Tropical and subtropical dry forests (hereafter: dry forests or TDF) occur on all continents and are among the most threatened ecosystems globally [1,2]. Many TDF are deforestation hotspots, due to the expansion and intensification of different forms of agriculture and forest use [3]. At the same time, TDF remain weakly protected in many regions [1]. TDF harbour unique biodiversity, including a great number of endemic taxa [4–6], and support the livelihood of many millions of people [7,8]. Understanding patterns, drivers and outcomes of forest changes in TDF is therefore important. Despite this, TDF have received far less attention than tropical rainforests. Available studies have mostly focused

on agricultural expansion and deforestation, while the status of those forests that are spared from the conversion is unclear.

Processes related to forest degradation remain weakly understood, although forest degradation likely affects large tracts of TDF [9]. The consequences of forest degradation are also significant. For example, forest degradation contributes in major ways to carbon emissions [10–12] and increases forests' susceptibility to fires and droughts [13,14]. Fires also exacerbate the impact of logging and fragmentation on biodiversity, and can eventually lead to a shift in forest state, including: a simplification of forest structure; domination by shrubs and pioneer species; loss of important ecological functions; or an increase in invasive species [15]. Better monitoring of forest degradation in TDF is thus key to improving our understanding of their status and threats, and to informing conservation and land-use planning.

A variety of disturbances have an impact on TDF and can be linked to degradation. These disturbances include both natural ones (e.g., drought, fires, storm events, floods) and anthropogenic ones (e.g., selective logging of valuable timber, logging for fuelwood collection or charcoal, mining, forest grazing, shifting cultivation) [1,16–18]. Disturbance agents can be natural processes or anthropogenic activities (e.g., natural fires vs. fire used as a management tool). Depending on the disturbance characteristics, different post-disturbance development trajectories can unfold, from full recovery to degradation cascades when disturbance frequency is high [19]. Thus, to understand forest changes and their impacts, it is important to identify and attribute disturbance agents to mapped areas of forest disturbances.

The recent rapid developments in satellite image access, algorithms, and computing power now allow to map forest disturbance at high spatial and temporal resolutions, and across large extents [20]. Although most applications have so far focused on boreal, temperate or tropical wet forest ecosystems, the mapping of forest disturbances in TDF has only recently received attention [21–26]. The attribution of disturbance agents remains an open research frontier—true for TDF and forest disturbance, generally [27,28]. A promising approach is to derive spectral-temporal metrics from Landsat time series segmentation, and then use these metrics in machine-learning algorithms to identify and map disturbance types or agents [29–33]. For example, the spectral magnitude, duration, rate of change, or spectral values before and after the disturbance can all help differentiate disturbance agents, such as fires, selective logging, clear-fell, storms, or insects [29,32,34,35]. Likewise, metrics describing the spectral recovery can be informative [36]. For example, the gap caused by a single felled tree, or single skid trail, might disappear quickly, while the scar of a forest fire might take decades to fade. The segmentation algorithm, LandTrendr [37], has been extensively tested for describing such spectral-temporal characteristics of disturbances [38–42]. It is implemented in Google Earth Engine, allowing for its wide application [43]. However, the capability of LandTrendr to identify disturbance agents in TDF has so far not been tested.

In addition to the spectral-temporal properties of disturbances, the spatial characteristics of disturbance patches provide an additional source of information for characterizing disturbance agents. Disturbance events, both natural and anthropogenic, typically affect areas larger than a single Landsat pixel, making the disturbance patch a useful unit to study [30,31]. Once disturbance patches are identified, their sizes and shapes can be derived to help distinguish agents [29,36]. For example, clear-cutting often results in geometrically shaped, large patches of tree loss; fire scars are large and irregularly shaped; and selective logging produces small disturbance patches. Once disturbances have been attributed to agents, analyses of their spatial determinants can provide insights into what drives disturbance regimes [44]. Most TDF have been inhabited and used over long periods of time and a variety of actors operate, use and shape these forests today [2,45,46]. For example, where forest fires occur predominantly inside large forest patches, they could be a part of natural disturbance dynamics or a result of indigenous and management practices, whereas fires adjacent to agricultural fields or settlements are likely of different origin [47]. Understand-

ing relationships between forest disturbance agents and spatial determinants thus helps to link disturbance patterns, agents and actor groups as a basis for disturbance management.

Here, we employed a methodology to detect and map disturbance agents in TDF based on satellite-based metrics. We applied this approach to the entire Argentine Dry Chaco, a vast region with a long history of forest use and degradation [48]. Forest conversion has been a key land change recently in the Argentine Dry Chaco, mainly due to the expansion of agribusinesses. However, the status of, and recent trends in forest condition of remaining, still-sizeable dry forests in this region, are unclear. Existing studies point to considerable forest degradation [49–51], as evidenced by woody cover decreasing close to fields, smallholder homesteads, and roads [52]. However, the prevalence and spatial patterns of the main disturbance agents have never been analyzed in the Chaco.

Building on a temporally- and spatially-detailed mapping of forest disturbance across the entire Argentine Dry Chaco from our own previous research [22], here, we aimed at identifying major disturbance agents in this system. Specifically, we combined multiple metrics describing disturbances spectrally and spatially into a random forests classifier to identify disturbance agents at the patch level. We then compared the identified disturbance agents to a range of features associated with key land-use actors' potential disturbance-driving patterns. Specifically, we asked:

1.　What was the prevalence of different disturbance agents in the period from 1990 to 2017 in the Argentine Dry Chaco?
2.　What were the dynamics of different types of forest disturbances in this time period?
3.　How do different disturbances' agents relate to anthropogenic features in the Chaco landscape, namely agricultural fields, forest smallholder homesteads, and roads?

## 2. Study Area

The Gran Chaco in South America is among the largest remaining continuous tropical dry forest in the world [53]. The majority of the ecoregion is located in northern Argentina, where it covers an area of 489,000 km$^2$. The Argentine Chaco stretches through a mostly flat terrain characterized by a strongly seasonal climate, with an average temperature of around 22 °C. The cooler, dry season is between May and September, and the hot, wet season is from November to April. Annual rainfall decreases from east to west from 1200 mm to 450 mm in the center of the region, with awestward increase closer to the Andes as a result of the orographic effect (Minetti 1999). Vegetation consists of a mosaic of xerophytic forests, open woodlands, scrubs, savannas and grasslands. Characteristic tree species belong to the generum *Schinopsis*, and in particular, *S. balansae* ("Quebracho colorado chaqueño"), *S. quebracho-colorado* ("Quebracho colorado"), *S. hankeana* ("Horco quebracho"). Moreover, Chaco forests include *Bulnesia sarmentoi* ("Palo santo"), *Aspidosperma quebracho-blanco* ("Quebracho blanco"), and *Prosopis* spp. ("Algarrobo"). The shrub layer of the Chaco is dominated by *Acacia*, *Mimosa*, *Prosopis* and *Celtis*, as well as the cacti *Opuntia* and *Cereus*. Forests sometimes intermix with natural grasslands and savannas, dominated by the grasses *Elionorus musitcus* or *Spartina argentinensis*. Finally, palm savannas with the palm *Copernicia alba* can occur in wetter parts of the Chaco [48,54,55].

Until recently, the Chaco was largely forested. Beginning in the 1980s, but especially in the 2000s, large-scale conversion of the Chaco's forests to agriculture occurred [56,57] leaving about 72% of forest cover today (for the Chaco as a whole) [58]. Most of the remaining Chaco forest is used by different actors, with many regions considered degraded due to historically unsustainable exploitation. This resulted in a substantial simplification of forest structure and composition, with a loss of trees in the upper layer, loss of the herb layer, or shrub encroachment [59]. Furthermore, the expansion of industrial agriculture also caused an intensification of forest use, worsening forest degradation [60]. Different land-use activities have contributed to the degradation of the Chaco forest. First, the tannin and wood industry has targeted valuable tree species, such as Quebracho Colorado, Palosanto and Algarrobo, leading to extractive logging for timber across wide swaths of the Argentine Chaco [59]. Second, much logging is linked to charcoal production, a

common economic activity of poorer rural people [49,61]. Third, the Chaco harbors many forest smallholders, locally referred to as '*puesteros*' or '*criollos*', who live inside the forest and use the surroundings of their homesteads for sustenance (e.g., fuelwood collection, timber for construction) [62]. The livestock of these smallholders typically roam freely around homesteads and have a considerable impact on forest structure [63]. Fourth, crop field and pasture management techniques—in areas where the forest was replaced by agriculture—include the use of fire, which can spread into the surrounding forest. The prevalence, frequency and spatial patterns of these anthropogenic disturbances, however, remain weakly understood at the regional scale. The same is true for natural disturbances, which include natural fires, droughts or flooding [55,64,65].

## 3. Materials and Methods

Our overall workflow (Figure 1) consisted of four steps. First, we derived spectral-temporal metrics at the pixel level from the temporal segmentation of the time series of Tasseled Cap Wetness and Normalized Burn Ratio composites. Second, we identified disturbance patches from a pixel-level disturbance map. Based on these patches, we calculated shape metrics and summarized the spectral-temporal metrics per patch. Third, we used these metrics in a random forest classifier to attribute disturbance agents to each disturbance patch. Finally, we assessed the spatial relationships of disturbance agents with anthropogenic features in the Chaco landscape, specifically agricultural fields, forest-smallholders, homesteads, and roads.

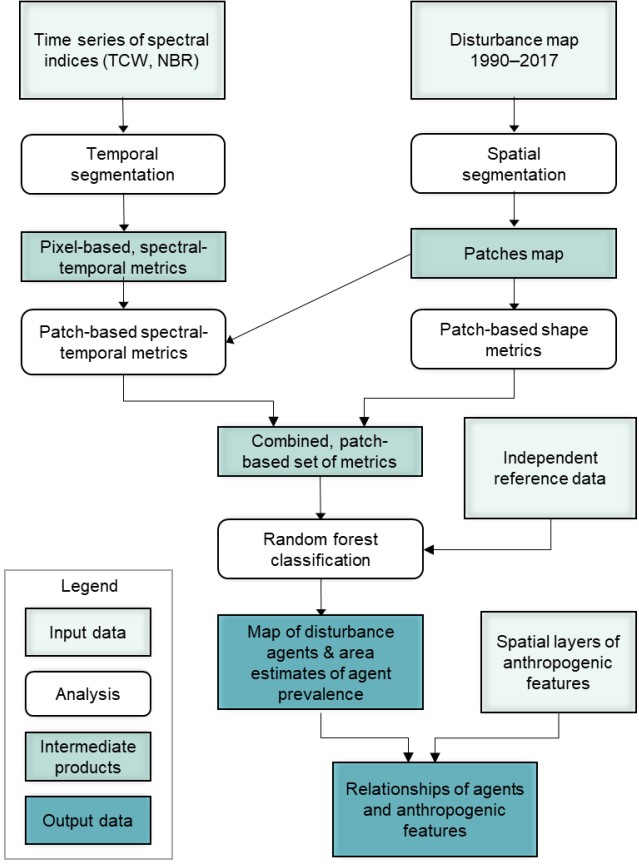

**Figure 1.** Overall workflow of the research methodology employed to identify and map disturbance agents in the Argentine Dry Chaco.

### 3.1. Forest Disturbance Map

We used a detailed forest disturbance map (Figure 2) for the Argentine Dry Chaco for the period 1990–2017 at 30 m resolution [22]. The map was produced using all available Landsat imagery and a time series change detection methodology. Specifically, our

methodology was based on calculating annual image composites of a set of spectral indices (i.e., Tasseled Cap Wetness—TCW, the Normalized Burn Ratio—NBR, and the Normalized Difference Moisture Index—NDMI), and then using trajectory analyses per pixel to identify disturbance years, as well as disturbance metrics. We then used an ensemble classification across these composites to derive a consensus disturbance map for all areas that were forested in 2017. The final map showing the location and timing of forest disturbances was rigorously validated following best-practice accuracy assessment protocols, using independent reference data. This disturbance map had an overall accuracy of about 79% and a user's accuracy of the disturbance class of 73% [22]. This map focuses on forest disturbances, not deforestation. In other words, we investigated forest loss that did not result in a change in land use. All deforested areas (i.e., forest areas cleared and converted to agricultural land use) were masked.

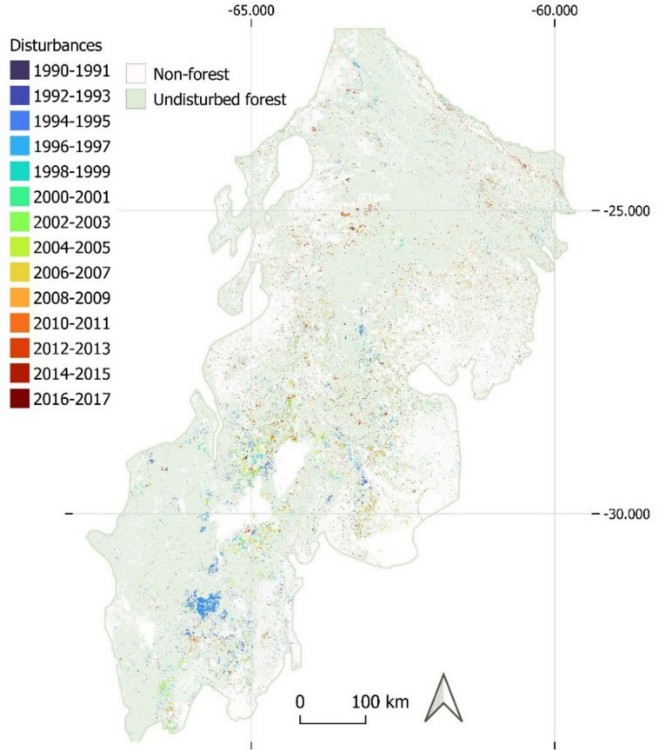

**Figure 2.** Disturbance map for the Argentine Dry Chaco showing the timing of individual disturbed pixels.

### 3.2. Spectral-Temporal Metrics

As a first step to further characterize disturbances, we calculated spectral-temporal metrics at the pixel level using the temporal segmentation algorithm LandTrendr [37]. LandTrendr fits simplified trajectories by first iteratively identifying a set of vertices (i.e., breakpoints) in the time series of a spectral index, and then fitting linear segments between these vertices (see [37] for details). Consistent with our prior work, and to match the forest disturbance maps described above, we derived time series' from all available Landsat TM, ETM+ and OLI images for the period 1987–2017 as Collection 1 Tier 1 surface reflectance data. Preprocessing consisted of clouds and cloud shadow masking and harmonization of surface reflectance values between OLI and ETM+ sensors [66]. For a more detailed description of the data and preprocessing, we refer to [22]. We did this based on annual Tasseled Cap Wetness composites, as it was the single best-performing index for disturbance detection in the Chaco in our prior work [22] and has been used for discriminating disturbance agents elsewhere [31,34]. From the resulting fitted segments, a number of metrics that characterize change over time can be derived—e.g., the length and numbers of

segments, spectral values at the beginning and end of segments, values and signs of the spectral change along with segments. We derived those metrics using the same settings and parameters used to produce the forest disturbance map [22]. In other words, we used the same segments and timing of disturbance to ensure agreement on the datasets. Individual disturbance types might be better captured by different spectral indices, and using more than one index could therefore improve disturbance agent attribution. Specifically, the Normalized Burn Ratio (NBR) could be useful for attributing fire as an agent. We therefore also extracted NBR values for our disturbances and pre- and post-disturbance segments. We did this based on the TCW LandTrendr fitting procedure to obtain a comparable dataset (i.e., a fitted time series with vertices matching the vertex timing identified from TCW time-series segmentation), with two spectral dimensions (TCW and NBR), as suggested by Kennedy et al. [30]. From these fitted time series, we derived metrics describing the state prior to the disturbance (i.e., spectral value in the year before the disturbance), the disturbance magnitude and duration, the state after the disturbance (i.e., post-disturbance spectral value), and the magnitude and duration of the post-disturbance segment (i.e., indicating recovery trajectories).

### 3.3. Identifying and Characterizing Disturbance Patches

To identify disturbance patches from our pixel-based disturbance map [22] we aggregated spatially adjacent, disturbed pixels into patches using an 8-neighbor adjacency rule and a minimum mapping unit of 1 ha (11 pixels). We assume in all subsequent analyses, that adjacent disturbance pixels can be attributed to the same disturbance agent. Based on these patches, we then calculated the following spatial metrics per patch in Google Earth Engine: patch area, patch perimeter, perimeter-area ratio, and the fractal dimension index. The latter reflects shape complexity across a range of spatial scales and was calculated as described in [67]. Next, we summarized the spectral-temporal predictor variables per disturbance patch by calculating the average and standard deviation metric values, yielding a patch-based dataset of 22 metrics (18 spectral-temporal and 4 spatial metrics; Table 1).

**Table 1.** Predictor variables calculated for each disturbance patch used for random forests modeling of disturbance agents.

| Patch-Based Metric | Variable (# Metrics) | Description |
|---|---|---|
| Spectral-temporal metrics | | |
| Pre-disturbance | Prevalue (2) | Mean of the spectral value before the disturbance of Tasseled Cap Wetness (TCW) and Normalized Burn Ratio (NBR) |
| Disturbance | Magnitude (4) | Mean and STDV of the spectral magnitude (difference between spectral values at the end and beginning of the disturbance segment) of TCW and NBR |
| | Relative magnitude (2) | Mean of the ratio between Magnitude and Prevalue TCW and NBR |
| | Duration (1) | Mean of the duration in years of the disturbance segment (same for TCW and NBR time series) |
| Post-disturbance | Endvalue (2) | Mean of the spectral value at the end of the disturbance of TCW and NBR |
| Recovery | Magnitude (4) | Mean and STDV of the difference between spectral values at the end and beginning of the recovery segment TCW and NBR |
| | Duration (1) | Mean of the duration in years of the recovery segment (same for TCW and NBR time series) |
| Spatial metrics | | |
| | Area (1) | Patch area |
| | Perimeter (1) | Patch perimeter |
| | Perimeter/area (1) | Ratio between patch perimeter and area |
| | Fractal index (1) | Patch fractal index |

### 3.4. Disturbance Attribution

To attribute each disturbance patch to its disturbance agent, we used a random forests classification. Based on the literature on forest disturbance and degradation in the Chaco, our extensive field knowledge from the region, as well as an initial scoping exercise where we examined disturbance patches in high-resolution imagery on Google Earth, we sought to attribute disturbance patches to one out of five possible agents: (1) logging, (2) fire, (3) partial clearing, (4) riparian changes, and (5) drought. To train our attribution algorithm, we collected 308 training patches across the study area (i.e., 70 logging, 70 fire, 77 partial clearing, 40 riparian changes, and 51 drought patches), where disturbance agents had been observed in the field and/or where disturbance agents could be clearly identified in Google Earth imagery.

"Logging" included selective logging for timber and logging for fuelwood collection or charcoal production. These activities typically leave a characteristic signature of a maze of irregular skid trails inside the forests, which can be easily identified on high-resolution imagery (Figure 3). The "fire" category included any fire (natural or anthropogenic) occurring inside forests. Fires are easily recognizable by the irregularly shaped patches, often an elongated shape in the north-south direction (due to the prevailing wind directions in the Chaco), and the spectrally-distinct signal of burned areas in the year of the fire. Our agent class "partial clearing" contained areas of forest where part of the canopy was removed as agriculture expanded. Partial clearing can occur because: (1) the conversion from forest to an agricultural field or pasture was initiated but never completed; (2) forests were converted but then abandoned, with subsequent forest regrowth; or (3) forest was converted to silvopastures, where some of the canopies are left on pastures. The agent class "riparian" refers to situations where the meandering of rivers in the Chaco (i.e., the Salado, Dulce, Pilcomayo and Bermejo rivers) leads to the erosion of riverbanks and therefore the degradation of riparian forests found on them. Given the highly dynamic fluvial systems in the Chaco, this can be common [68]. Finally, our agent "drought" included areas where a severe rainfall deficit leads to a disturbance signal, as the vitality and productivity of the canopy are reduced in the drought and subsequent years.

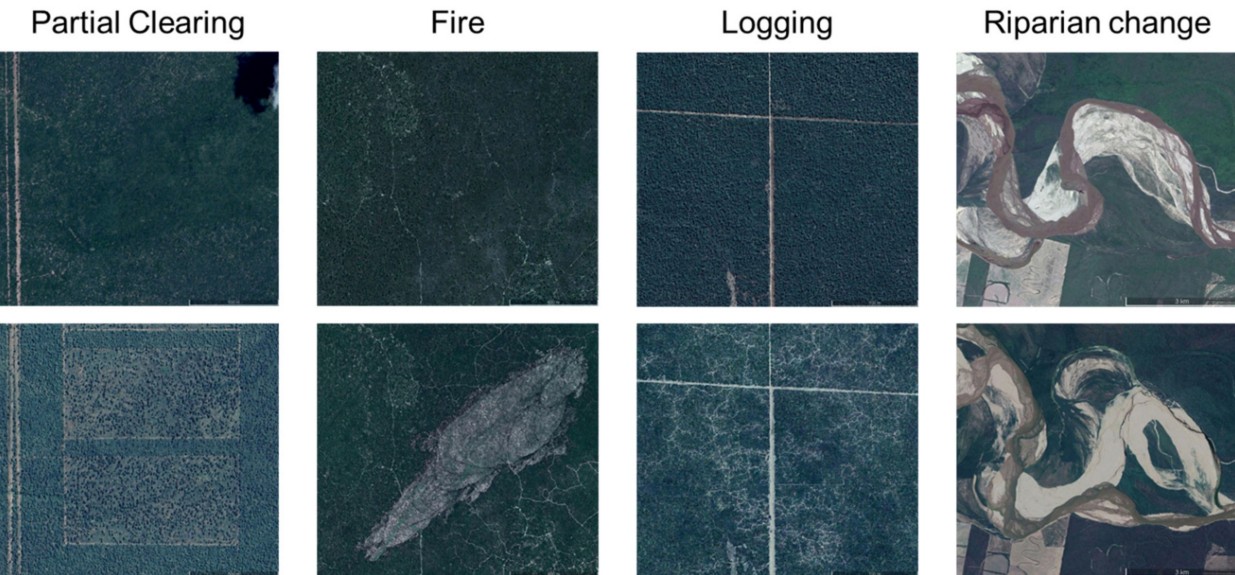

**Figure 3.** Examples of different disturbance agents as seen on very-high-resolution images from Google Earth. **Top row**: forests prior to the disturbance. **Bottom row**: post-disturbance situations. Columns: Partial clearing (in the example a silvopasture field); fire; logging; riparian change (in the example, the Bermejo river. Note how mutable the river meanders are).

We used our set of patch-level metrics together with our training data to train a random forests classifier (RF, [69]) to assign disturbance agents per patch. Random forests are a non-parametric classifier that consists of many individual decision trees that together determine class attribution. Individual decision trees are built using a bootstrap aggregation technique (i.e., bagging), which randomly samples a subset from the full training dataset to derive a tree, resulting in a 'forest' of many different decision trees. This has the advantage of reducing overfitting problems. Random forests classifiers are a powerful tool for remote sensing classifications generally, and particularly for attributing disturbance agents [29–31]. Here, we applied the algorithm at the patch level, assigning one of the five agents to each patch. Our classification was based on 500 decision trees.

We validated our resulting disturbance agent map with a pixel-based accuracy assessment. We used a two-stage sampling strategy. First, we randomly sampled patches from a list of all patches, stratified by disturbance agent. Because our disturbance agents resulted in differently sized patches (e.g., the fire had typically large patches), this ensured representation of all agent types and a diversity of patch sizes in our sample. Moreover, reliable labeling of disturbed pixels to agents requires considering the patch context (i.e., size and shape of patches) and sampling patches randomly as a first step, thus avoiding bias. Second, we further sampled 70 pixels per stratum with a minimum distance of 100 m between them. We then inspected each pixel visually on-screen in (1) high-resolution Google Earth imagery, and (2) Landsat images' time series of raw spectral and TCW using the Time series app in Google Earth Engine (https://github.com/jdbcode/ee-rgb-timeseries, accessed on 11 August 2021). We labeled the disturbance agent corresponding to the disturbance that occurred in the year indicated by our disturbance map [22]; in other words, in case of multiple disturbances per pixel, we assigned the agent based on the year of detection (i.e., dominant disturbance) and in case of multi-year disturbances we assigned the dominant year for the entire patch. This allowed to assign a disturbance agent to each pixel. In total, we could label 326 pixels. We then used this reference dataset to calculate a confusion matrix, from which we estimated area estimates as well as overall and class-wise accuracies, using the estimator for stratified random sampling, following best-practice protocols [70,71].

### 3.5. Analysing Disturbances in Relation to Agricultural Fields, Homesteads and Roads

To further understand the distribution of disturbance agents in space and time, we carried out a number of geospatial analyses. First, we analysed the share of disturbed areas attributed to our different disturbance agents over time, based on our disturbance agent map. This resulted in trend graphs per agent. Second, we compared our disturbances to land-cover maps (Table 2) to assess whether some disturbance types were more common close to agricultural fields. As agriculture has been expanding rapidly into forests in the Argentine Chaco over our observation period, this required us to match the timing of each disturbance with an agricultural map from that time period. To do so, we used a time series of land-cover maps in 5-year intervals (1990, 1995, 2000, 2005, 2010, and 2015) based on Landsat satellite imagery [72]. We then calculated 500 m buffers up to a maximum distance of 4000 m around fields and summarized the area of disturbance patches by agent and buffer. This allowed to assess the distribution of disturbance agents relative to the distance to agricultural land.

Third, we compared our disturbance agent map to a dataset of locations of forest smallholder homesteads (Table 2), obtained by screen-digitalizing about 24,000 individual homesteads from Landsat and Google Earth with very-high-resolution imagery for our entire study region [62]. This database includes temporal information, specifically on which homesteads were persistent over the time period 1985–2015, and which homesteads disappeared or emerged (and when). Using the same procedure as above, we summarized the areas of disturbances by agent around buffers surrounding homesteads (i.e., considering which homesteads were present when the disturbance had taken place).

Finally, we analyzed disturbance agents relative to the distance to roads (Table 2). We used a time series of road networks for the years 1995, 2000, 2005, 2010, 2015, re-

constructed from OpenStreetMap data, historical road atlases, and historical imagery on Google Earth. Our dataset contained all major paved and unpaved roads. We then derived areas of disturbances by an agent in relation to distance, again considering when roads were constructed.

**Table 2.** Anthropogenic features in the Chaco landscape used for the analysis of the spatial distribution of disturbance agents.

| Variable | Source | Reference |
| --- | --- | --- |
| Distance to agricultural fields | Land-cover maps for the years 1990, 1995, 2000, 2005, 2010, and 2015 | [72] |
| Distance to smallholders homesteads | Homesteads screen digitalization based on the Landsat archive and very-high-resolution imagery in Google Earth | [62] |
| Distance to roads | Road network for the years 1995, 2000, 2005, 2010, 2015 | openstreetmap.org (accessed on 15 May 2017), [73] |

## 4. Results

### 4.1. Prevalence and Estimated Areas of Different Disturbance Agents

Our best random forests model of disturbance agents had an area-adjusted overall accuracy of 56.6%. Class-wise user's accuracies were highest for partial clearing (85.5%), followed by fire (64.3%), whereas the user's accuracy was lower for drought and logging (60.9% and 36.5%, respectively). Riparian changes had the lowest accuracies (20.0%). In terms of producer's accuracy, the class fire was most reliable (70.6%), although there was some confusion, particularly with logging. The partial clearing had a producer's accuracy of 48.9%, with pixels misclassified with all other classes (Table 3). Logging pixels were mostly misclassified with riparian and fire. For the drought class, a number of reference pixels were attributed to logging in our classification. The riparian changes class had, again, the lowest accuracy (38.5%). Generally, our accuracy assessment revealed a fairly even error distribution (i.e., between user's and producer's accuracies). We used the model to generate our disturbance agent map (Figure 4).

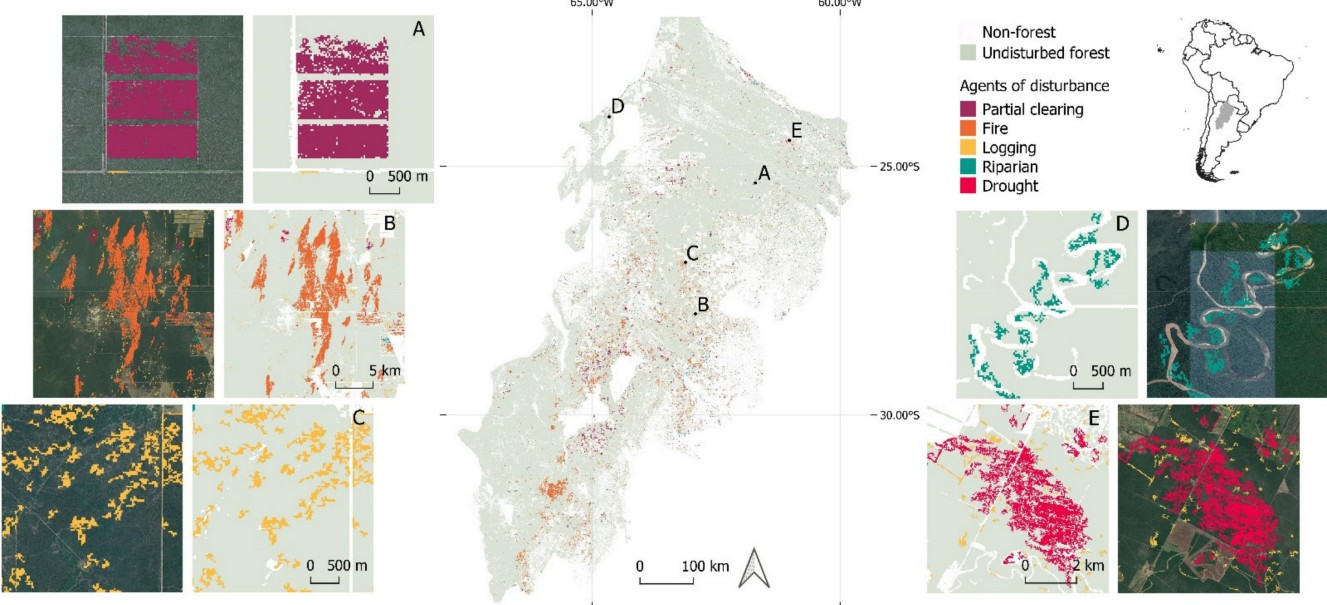

**Figure 4.** Map of forest disturbance agents for the Argentine Dry Chaco. Insets show examples for the five disturbance agents identified in our analyses: (**A**) Partial clearing, (**B**) Fire, (**C**) Logging, (**D**) Riparian change, (**E**) Dorught (Google Earth Imagery as background).

**Table 3.** Population error matrix showing the estimated percentage of pixels attributed to each agent class in the reference (columns) and by the model predictions (rows), as well as the estimated user's and producer's accuracies per disturbance agent class with the relative standard error.

|  |  | Observed | | | | | |
|---|---|---|---|---|---|---|---|
|  |  | **Partial Clearing** | **Fire** | **Logging** | **Riparian** | **Drought** | **User's Accuracy** |
| **Predicted** | **Partial Clearing** | 16.1 | 1.4 | 1.1 | 0.0 | 0.3 | 85.5 (±4.3) |
|  | **Fire** | 5.4 | 21.9 | 4.9 | 0.0 | 1.9 | 64.3 (±5.8) |
|  | **Logging** | 6.9 | 6.0 | 10.6 | 1.8 | 3.7 | 36.5 (±6.1) |
|  | **Riparian** | 2.2 | 1.1 | 1.8 | 1.5 | 0.7 | 20.0 (±5.2) |
|  | **Drought** | 2.3 | 0.7 | 0.7 | 0.5 | 6.5 | 60.9 (±6.1) |
|  | **Producer's Accuracy** | 48.9 (±3.6) | 70.6 (±4.2) | 55.6 (±6.3) | 38.5 (±11.3) | 49.4 (±6.5) |  |

Although our classification of disturbance agents contained, as can be expected, considerable uncertainty, it is important to note that our area estimates are associated with narrow confidence intervals (Figure 5). These area estimates, derived using independent reference data, revealed that partial clearing, a disturbance associated with the wave of agricultural expansion in the region, was the most widespread disturbance agent (Figure 5), covering $5476 \pm 789$ km$^2$. The second most widespread agent was fire ($5171 \pm 843$ km$^2$) followed by logging ($3176 \pm 785$ km$^2$). Smaller areas were affected by drought ($2189 \pm 566$ km$^2$) and the least widespread disturbance agent was riparian changes ($635 \pm 170$ km$^2$). In total, disturbances that can be clearly attributed to natural causes (i.e., drought and riparian changes) accounted for 17% of the disturbed area ($2824 \pm 898$ km$^2$), whereas disturbance agents that can be clearly attributed to anthropogenic activities (i.e., partial clearing and logging) accounted for 52% of the disturbed area ($8652 \pm 1574$ km$^2$). Note that fires could be both natural or of anthropogenic origin.

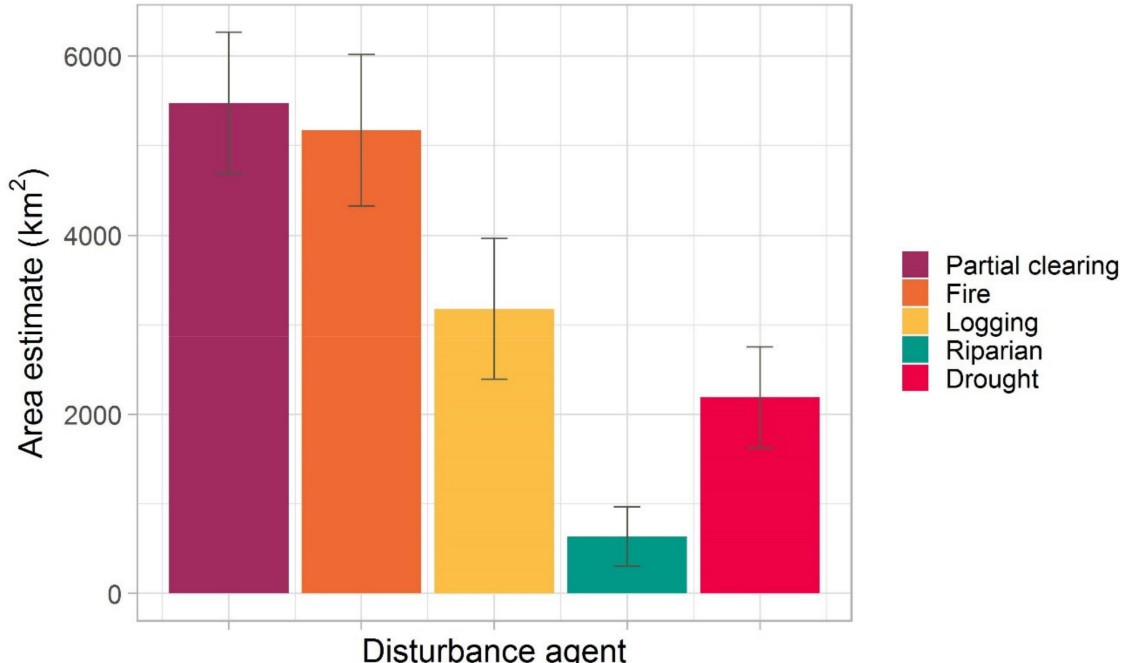

**Figure 5.** Area estimates including uncertainty range (i.e., confidence intervals) for five major forest disturbance agents for the Argentine Chaco for the period 1990 to 2017.

### 4.2. Trends in Disturbance Agents

Summarizing disturbance agents over time showed interesting trends (Figure 6). Analyzing trends in absolute agent distribution (i.e., based on map estimates, Figure 6A) revealed that partial clearing, a disturbance connected to agricultural expansion, was overall not very widespread (<70 km²/year) until the mid-2000s, when it abruptly increased two- to fourfold, with a maximum area of 285 km² in 2004. Logging was relatively widespread in the period 1992–1997 (around 190 km²/year), then lower in 1998–2003, yet much more widespread between 2004 and 2013 (around 330 km²/year). The highest area of logging occurred in 2013. Areas affected by fire fluctuated more heavily, with peaks around 1995 (1090 km² in 1995, 666 km² in 1996) and 2004 (608 km² in 2004 and 420 km² in 2005). Minimum fire years were 1990 (only 1 km²) and 2015–2017 (3–9 km²). Drought was overall a not very widespread disturbance agent, with drought-affected areas particularly prevalent in 1993, 2000, 2005, and 2012–2013.

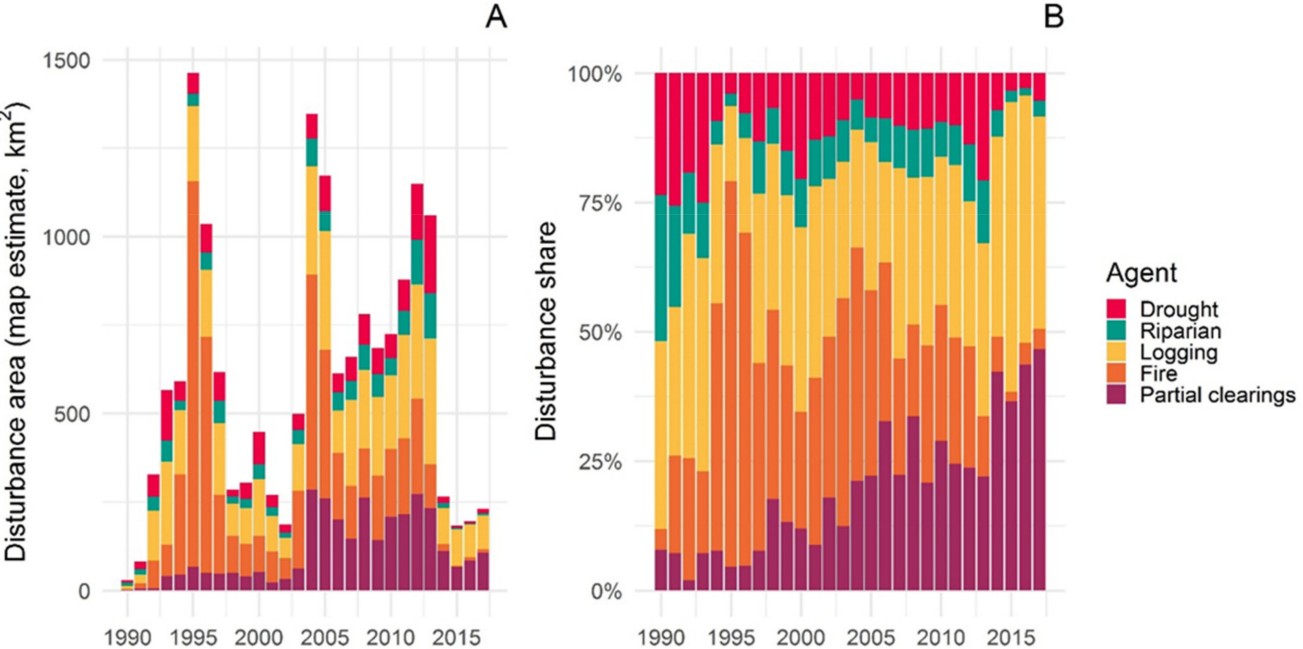

**Figure 6.** (**A**) Disturbance areas by year, and (**B**) areas proportion of the five agents by year.

Assessing the relative share of disturbance agents over time showed clear trends in disturbance importance (Figure 6B). Most importantly, the share of partial clearing steadily increased, with shares rising from <10% in the early 1990s to >40% after 2010. The share of fire decreased over time in contrast, yet with major fluctuations over our observation period. Fire-prone periods were from 1994 to 1996 (between 47% to 74%), 2003–2004 (around 45%) and 2009–2010 (around 26%), whereas fire became less important in the Argentine Dry Chaco after 2015. Logging prevalence stayed relatively stable, at around 30%, although the area of Chaco forest declined substantially over our observation period. According to our map, drought prevalence was higher in the early years of our study period, as well as during 2000–2013.

### 4.3. Disturbance Agents in Relation to Anthropogenic Features

Analyzing the spatial patterns of our disturbance agents in relation to anthropogenic features, specifically agricultural fields, smallholder homesteads and roads, revealed interesting trends away from these features (Figure 7). Trends were overall fairly similar across disturbance agents (columns in Figure 7) and within specific anthropogenic features (rows in Figure 7). In terms of agricultural fields, we found the clearest patterns, with all disturbances declining away from fields. For partial clearing and fire, we found an initial increase

away from fields, with peaks of these disturbances about 1 km away from the fields. These patterns were different when analyzing disturbances in relation to smallholder homesteads. Here, we found an initial increase in disturbances, particularly for partial clearing and fire, with a peak of around 1–2 km, but a much more gradual decline further away. These patterns were similar for partial clearing, fire and logging, which, however, differ from the patterns found for riparian changes and drought (less decline away from smallholder homesteads than for the other agents, middle row). For roads, we found similar patterns for smallholder homesteads, with an initial peak at 1–2 km and a subsequent gradual decline. Again, this pattern was clearer for anthropogenic compared to natural disturbance agents (Figure 7).

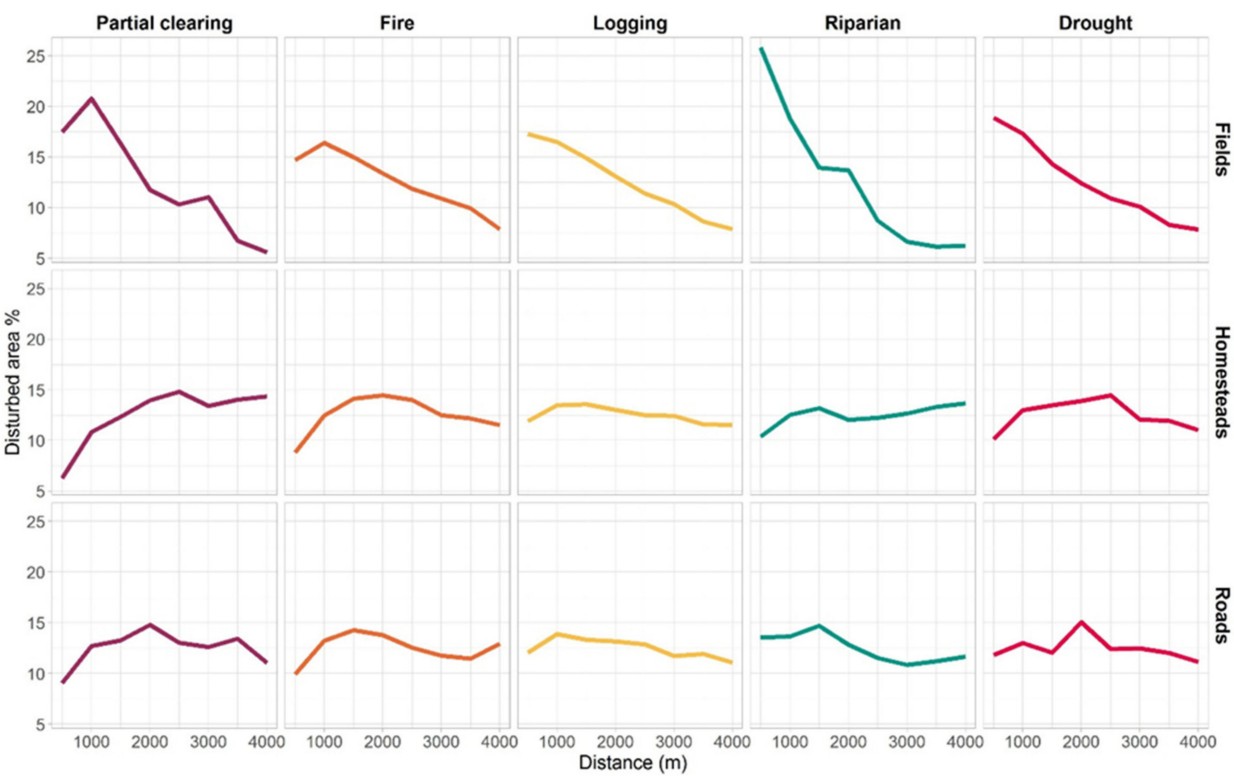

**Figure 7.** Disturbance agents in relation to distance to agricultural fields, smallholders' homesteads and roads.

## 5. Discussion

Understanding the agents of forest disturbances is important for avoiding and addressing forest degradation, biodiversity loss and climate change. This is particularly true for the world's widespread, often threatened, but frequently neglected tropical and subtropical dry forests. Using Landsat time series and a patch-based classification framework, we here reconstruct the prevalence and dynamics of five forest disturbance agents over a 30-year time span across the entire Argentine Dry Chaco, a vast dry forest region (489,000 km$^2$) and global deforestation hotspot. This suggested that multiple, co-occurring disturbance agents contribute to forest degradation trends in the Chaco. Specifically, our main findings were, first, that partial clearing was the most widespread disturbance agent, pointing to a so far largely overlooked outcome of agricultural expansion processes. Second, fire and logging affected sizeable areas of the remaining forests, with most fires likely of anthropogenic origin, pointing to an urgent need for fire management strategies to preserve the remaining Dry Chaco forest. Third, most disturbances, particularly fire and logging, decreased markedly away from agricultural areas (i.e., crop fields and pastures), highlighting so far undocumented edge effects; an indirect outcome of the recent wave of agricultural expansion. Finally, we demonstrate a considerable impact of forest smallholders on dis-

turbance prevalence, but also that this impact is spatially restricted to the vicinity of their farms. More generally, our study highlights that an improved degradation monitoring of disturbance agents is needed to sustainably manage dry forests, in the Chaco and elsewhere. Our approach, based on readily available Landsat archives, is promising for improving the understanding of the links between disturbance patterns and actors.

Partial clearing was the most widespread disturbance agent we identified, and the prevalence of partial disturbances increased, particularly after 2000. Two factors can explain this finding. First, partial clearing likely includes forests that were initially cleared for agriculture but later abandoned, allowing the regrowth of a secondary forest. Initial clearing may be carried out to ensure land claims, yet sometimes farmers might not be able to afford to establish agriculture. Subsequently, agricultural operations fail, or the initial clearing is carried out for speculation purposes (i.e., to re-sell cleared land). The Argentine Forest Law (Law 26.331, Ministerio de Agricultura Ganadería y Pesca, 2015) was discussed and finally implemented in 2007, enacting considerable land use restrictions over large areas of forests. Many landowners might have converted forests in fear of not being able to do so later, which likely led to many situations where land was not put to use due to the reasons outlined above. Second, silvopastures, where part of the canopy is retained on pastures, became widespread after the passing of the Forest Law, as this land use was still allowed in areas where full conversion became prohibited. This incentivized silvopastures [74], which is likely captured in our partial clearing class. Generally, the strong increase in partial clearings we found for the 2000s corresponds well with other findings [57,72] that highlight an acceleration of forest loss due to the agriculture boom in this period.

Fire was the second most important disturbance agent. Although natural fires can occur, most of the fires in the Argentine Chaco result from human ignition sources [65,75]. Fire is used for vegetation clearing and as a management tool to control shrub encroachment and promote grass growth in pastures [76,77]. Where fire is used for management, fire can easily spread into neighboring forests—especially in dry years—and affect very large areas [78]. Indeed, some of the largest patches in our disturbance map were due to fires (i.e., of the ten largest patches, eight were attributed to fires, with areas ranging between 35 $km^2$ and 660 $km^2$). A link between fire and drought has been suggested by prior studies for the Chaco [78,79] and South America more generally [80], and indeed the two peak fire years we found in our analysis were also years of documented severe drought (1995 and 2004, Figure 6). Interestingly, fire seems to be a better way to map drought impact than our drought class, as the extremely dry years in 1995 and 2004 were captured in our time series of fire-affected areas, but less so in our drought-affected areas. Overall, though, fire prevalence in Chaco's forests decreased over time, which might represent higher levels of control over fires, particularly in areas where cropping takes place and farmers are investing in fire prevention/restriction [80]. Likewise, declining fire can be explained by a less (full) conversion of forests to agriculture, as fire has historically been used for clearing land, but this is no longer compatible with land-use restrictions that require keeping part of the canopy.

Logging was the third most widespread disturbance type we found, with a fairly constant share of forest affected over our study period. Logging can be related to the harvesting of valuable timber, production of firewood, fence poles, tannin or charcoal and these activities relate to different actors ranging from wood industries to smallholders using the selling of forest products as a safety net during economically hard times [61]. The large area affected by this disturbance type, and the fact that logging activities remained important despite major forest losses and an 'agriculturisation' of the region, highlights the importance of considering logging in assessments of forest integrity, pressure on biodiversity, or emission assessments—which has so far not been possible due to a lack of spatially-detailed and area-wide maps. We note that our estimates of logging are likely conservative, as logging normally affects smaller patches and we had a minimum mapping unit of 1 ha.

Our disturbance attribution was less reliable for the natural disturbances of riparian change and drought. Disturbances due to meandering rivers are important but are locally restricted phenomena, and therefore do not cover a large area. Integrating our disturbance analyses with an assessment of changes in water surface would likely increase the reliability of this class in major ways. Drought-related disturbance also affects a smaller area, however, it must be noted that our analysis likely does not capture all drought-related vegetation stress on Chaco forests. Mapping drought impact was not our main objective, and we can therefore see this class mainly as separating out strong drought impacts to avoid confusion with other disturbance agents. A more complete assessment of drought impact on vegetation would likely benefit from a temporally more resolved time series (e.g., MODIS time series). Interestingly, many of our mapped drought disturbances were found in paleochannels, indicating that vegetation occurring on these sandy soils is likely more sensitive to drought.

A striking pattern we observed, however, was the decreasing disturbance prevalence away from agriculture. For partial clearing, this is an expected trend, as distance to prior deforestation has the highest explanatory power for new land clearance [81]. The distinct initial increase in partial clearing prevalence away from fields is likely related to the average distance between fields, and in between there is still some forest in the form of narrow forest stripes left to prevent wind erosion ("cortinas"), which are often degraded (but overall cover a small area). Fire, as well as logging, decrease strongly away from fields, particularly beyond 1 km. This likely reflects the accessibility of forest, relevant for extractive activities but also as a predictor of human activities and thus human ignition. Furthermore, where fires are used for management purposes, they can escape into adjacent forests, as highlighted above.

We also found marked effects of smallholder homesteads in relation to disturbance prevalence. The decreasing partial clearing closer to smallholder homesteads likely indicates that homesteads persist only if they have sizeable forests in the surrounding, with many homesteads abandoned as industrialized agriculture expands around them [62]. Fire occurrence was lower closer to homesteads; on the one hand, because of higher fire control; on the other hand, likely because livestock reduces woody cover and herbaceous fuel loads in the close surroundings of homesteads [51,52]. This might also explain the increase in logging further away from homesteads, as wood availability should increase further away. We note that we found similar, though somewhat less conclusive, patterns for distances to roads in line with prior work [52], which can be explained by roads being a less clear indicator of human presence than settlements. Yet, this might also point to shortcomings in our road dataset, which was perhaps too coarse (mainly paved roads only) and did not include small roads opened for the purpose of logging or (historically) oil prospecting [77]. Our findings on the relationship of disturbances to anthropogenic features are based on an analysis of the whole area, while different contexts might result in local differences. For example, the collection of firewood is influenced by household income and access to forestland and therefore varies in different regions [61]. Charcoal ovens are concentrated in some provinces and rare in others [49]. Pastures, where a fire is used as a management tool, prevail over crops in drier areas [82]. These factors might produce different local patterns of disturbances prevalence in relation to anthropogenic features.

Our analyses yielded robust area estimates and maps of disturbance agents, yet also highlighted the challenges of accurately attributing disturbance agents. The level of accuracy we achieved was comparable to other studies. For example, we obtained user accuracies of 77.0% for disturbances caused by partial clearing and 46% for our logging class, comparable to a harvest disturbance accuracy of 68% for Central Europe [27], or between 63% and 87% for the USA [36]. Our fire class had a user's accuracy of 59.3%, which is lower than what is often reported from other biomes (i.e., the Boreal), although fire mapping is easier there [83]. Interestingly, we did not find any study that provides robust error estimates for disturbance agents in dry forests, limiting comparability, and we found no study at all independently evaluating the performance of riparian change disturbances

or drought (typically included in classes like "other" [29] or "stress" [36], with widely varying error estimates, such as 29%−88% in the latter study). These then highlight the complexity of disturbances' agents attribution in tropical dry forests, and the urgent need for more studies in this biome. In our case, confusion was highest among partial clearing, fire and logging, disturbances that in reality, blend on the ground as they are all connected to both the deforestation process and to management (e.g., silvopastures, charcoal production). Better understanding agent complexes (i.e., co-occurring or sequential disturbances) would therefore be a useful next step. A deeper consideration of landscape context [27] could help in this regard, in addition to the spectral-temporal and patch shape metrics we used here.

Although our methodology resulted in a reliable disturbance agent attribution and robust area estimates for these agents across a large region, a few limitations need to be mentioned. First, we carried out the most comprehensive attribution of disturbance agents so far for the Chaco or any dry forest region, but we could not find reliable reference data for some disturbance types that are consequently not identified here. These might include disturbance due to salinization [84], insect disturbance, or herbicide drift due to heavy pesticide use on some crops (i.e., soybean, cotton). Second, we applied a minimum mapping unit of 11 pixels, equaling approximately 1 ha. This helped to remove scattered, small patches, many of which likely represent misclassification, but we cannot rule out that this also did not filter out some disturbance types connected to very small patches, such as logging, more than others, such as fire. Third, our disturbance map is likely a conservative estimate as we did not map low-severity disturbances [22] and our map does not capture sub-canopy disturbances, such as forest grazing and resulting forest understory degradation, a common process of forest degradation in the Chaco. Fourth, our disturbance map had by itself some level of uncertainty (see [22]) that are not fully captured in the accuracy metrics we report here, as these only captured the reliability of the agent attribution. However, our disturbance map had a very balanced omission and commission errors, and we, therefore, do not expect uncertainty in disturbance detection to strongly affect our disturbance agent area estimates.

## 6. Conclusions

Going beyond only mapping forest conversion in tropical and subtropical dry forests to more deeply consider forest disturbance and forest degradation, is urgently needed to better understand human pressure on these systems. Remote sensing is an essential tool for deforestation monitoring and should be a key tool for assessing disturbance and more subtle changes in these forests as well, but has so far not been widely used for this purpose. In this study, we demonstrated the benefit of the unique Landsat archive to assess maps, at a high spatial and temporal resolution, of different forest disturbances agents, and to separate anthropogenic from natural disturbances for the entire Argentine Dry Chaco. A number of studies focused on mapping or quantifying the conversion of forests to agriculture e.g., [56,57,72], and a few investigated the changes in the remaining forests, such as degradation [50], logging [49], and fires [79] in regions of the Chaco. However, this is, to our knowledge, the first forest disturbance agent attribution for the whole Argentine Dry Chaco. Given that our workflow is implemented in Google Earth Engine, there is considerable potential for consistent, repetitive forest disturbance monitoring, as well as for upscaling to larger areas—given that appropriate training and validation data can be gathered. Thus, our workflow can be a starting point for a monitoring tool supporting land managers, planners and policymakers. Thematically, our work suggests that a large proportion of the forest so far spared from deforestation is affected by anthropogenic disturbances, related to a diversity of land-use actors. This highlights the need to better capture and address forest degradation in order to maintain ecological integrity. Forest degradation, as an important group of processes, should not be neglected in tropical dry forests undergoing deforestation due to the expansion of commodity agriculture, such as in the Chaco.

**Author Contributions:** Conceptualization, data curation, formal analysis, methodology, writing—original draft, T.D.M.; conceptualization, writing—review and editing, N.I.G.; conceptualization, funding acquisition, supervision, writing—review and editing, E.F.L.; conceptualization, funding acquisition, supervision, writing—review and editing, T.K. All authors have read and agreed to the published version of the manuscript.

**Funding:** This research was funded by the Belgian Federal Science Policy Office Research Programme for Earth Observation (belspo-STEREO-III, project REFORCHA, SR/00/338).

**Institutional Review Board Statement:** Not applicable.

**Informed Consent Statement:** Not applicable.

**Data Availability Statement:** The data presented in this study are available upon request from the corresponding author.

**Acknowledgments:** We thank Rubén Chavez and Alfredo Romero-Muñoz for sharing the digitized road dataset, Christian Levers and co-authors for generating and sharing the smallholder dataset, Matthias Baumann and co-authors for sharing the land-cover maps.

**Conflicts of Interest:** The authors declare no conflict of interest.

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
