# Peer review of "Agents of Forest Disturbance in the Argentine Dry Chaco"

_remotesensing, doi:10.3390/rs14071758_

Round 1

Reviewer 1 Report

This study mapped forest disturbance agents using Landsat time series data in tropical dry forests in Argentine. The authors used disturbance maps developed in their previous study for agent attribution in random forest modeling based on predictor variables derived from LandTrendr temporal segmentation. They showed spatial and temporal patterns of these disturbance agents in the study area from 1990 to 2017.

Overall, this manuscript is well-written and easy to follow. The authors investigated an important topic in tropical dry forests and demonstrated the effectiveness of using Landsat time series data to obtain interesting results in this region. Although agent attribution has been implemented in several studies after first implemented in Kennedy et al. (2015), the application of this approach is still valuable, especially in the tropics. I feel the methods used are generally sound; however, there are several points that should be improved.

Major Comments:

- Although I agree that agent attribution has rarely been implemented in tropical dry forests, I do not find the novelty of the approach in this study. The authors stated that they “developed a methodology” in L104, but in my view, the methods used to attribute forest disturbance agents are quite similar to those used in previous studies (e.g. Kennedy et al. 2015). If there are some new aspects in their developed method, the authors should emphasize such points in the manuscript. Even if there is no original aspect, however, I assume the application of this approach is still valuable. In such a case, they should eliminate the confusing descriptions such as “developed”.

- The authors seem to focus only on forest disturbances that caused forest degradation and eliminate deforestation, in my understanding. After carefully reading De Marzo et al. (2021) and the definition of disturbance agents, the readers can understand that the disturbance agent map does not include converted areas from forest to other land use (i.e. deforestation); however, it is a little confusing. To avoid incorrect comprehension, I recommend the authors state that deforestation is not included and disturbance only related to forest degradation is used for agent attribution, maybe in the introduction.

-  It is good to calculate accuracy metrics based on unbiased estimators following the good practice recommendation; however, there are several unclarified points. The authors used polygon patches as spatial assessment units in the accuracy assessment; however, the detailed estimators are unclear in the manuscript. Because polygons have varying shape areas and different inclusion probabilities, the variance estimators in Olofsson et al. (2014) are not applicable to the polygon-based assessment. Although Radoux and Bogaert (2017) provide the good practices for the accuracy assessment of OBIA, variance estimators for Producer’s and User’s accuracies are not provided. Thus, it is unclear for the estimators that the authors used although they cite these publications. The authors are recommended providing the calculation methods of accuracy metrics in this study. Since the authors mainly focused on area estimates, using the pixel-based accuracy assessment might be an alternative choice. There seems to be no reason to assess polygon-based accuracy.

Specific Comments:

L72: Is tropical wet forest different from tropical rain forest?

L168: “Normalized Burnt Ratio” should be Normalized Burn Ratio.

L195: There is no description of Landsat data used in this study. I assume they used the same Landsat data in the same period in De Marzo et al. (2021); however, at least a brief explanation is required.

L200: I understand that LandTrendr segmentation of NBR has the same temporal segmentation as TCW segmentation using the fit-to-vertex procedure. I assume the timing of disturbances in the disturbance map from De Marzo et al. (2021) is also completely in accordance with the LandTrendr temporal segmentation in this study; otherwise, it is difficult to calculate patch-based metrics. In the current structure, however, such a relationship is ambiguous. I recommend clarifying that the LandTrendr segmentation in this study is the same as the previous study and the timing of disturbance is captured as segments.

L220: Patches were formed regardless of the disturbance year?

L229: Please correct the reference (to Table 1?).

L276: Although unbiased estimation is important for accuracy assessment, the calculation is unclear here. Polygons have the variable size of areas and thus have different inclusion probabilities. Thus, the variance estimators in Olofsson et al. (2014) “would not apply to a polygon assessment unit” (please see section 4.3). Radoux and Bogaert (2017) do not provide the variance estimators for producer’s and user’s accuracies. So, how did the authors calculate 95% confidence intervals of accuracy metrics? As far as I understand, due to such situations, most previous studies did not implement unbiased estimation based on objects, but simple sample count metrics. Because pixel-based accuracy assessment is still valid for assessing polygon-based maps when we assess the area-based metrics (please see Stehman and Wickham 2011), the use of pixels for accuracy assessment of agent attribution is another choice. Actually, several studies applied such protocols in agent attribution (e.g., Vogeler et al. 2020).

In addition, the authors are recommended to provide the response design of the assessment. In particular, it is better to provide how the agreement was defined in visual interpretation. In polygon-based maps, the perfect matching of disturbance shape in the map and ground rarely occurred; thus, the decision between maps and visual interpretation is necessary. Furthermore, there is no “no disturbance” class in the reference. Considering the UA of disturbance in the original dataset (although 11 pixels MMU might improve the UA), there might be disturbance patches that are actually not a disturbance. How did the authors cope with such patterns?

L321: “error-adjusted” or “area-adjusted”?

L321: The accuracy metrics here are applicable only for detected disturbance. Considering the PA and UA of forest disturbance in De Marzo et al. (2021), some parts of disturbance agents would be missed. Did the authors include such omissions in reporting areas estimates of disturbance agents? If not, I recommend the authors discuss the potential impacts of omissions of disturbance.

L334: It is better to provide a population error matrix in addition to a sample count based one.

L340: uncertainty bands mean 95% confidence intervals? Please clarify.

L382: Since disturbance occurs only in forests, the distribution of forests around fields, homesteads, and roads is important. The distribution of forests is almost the same for each category? If not, the distance relation might be affected by the distribution of forests.

L453–459: Does the description here mean that the reduction of fire might not be positive results (forest degradation is decreasing, but deforestation might be increasing)? The trend of deforestation in the same periods might be helpful to understand the situation in this study area.

L561: Although this is my preference, it might be better to include the conclusions section.

References:

  • De Marzo, Teresa, Dirk Pflugmacher, Matthias Baumann, Eric F. Lambin, Ignacio Gasparri, and Tobias Kuemmerle. 2021. “Characterizing Forest Disturbances across the Argentine Dry Chaco Based on Landsat Time Series.” International Journal of Applied Earth Observation and Geoinformation 98: 102310. doi:10.1016/j.jag.2021.102310.
  • Olofsson, Pontus, Giles M Foody, Martin Herold, Stephen V Stehman, Curtis E Woodcock, and Michael A Wulder. 2014. “Good Practices for Estimating Area and Assessing Accuracy of Land Change.” Remote Sensing of Environment 148: 42–57. doi:10.1016/j.rse.2014.02.015.
  • Radoux, Julien, and Patrick Bogaert. 2017. “Good Practices for Object-Based Accuracy Assessment.” Remote Sensing 9: 646. doi:10.3390/rs9070646.
  • Stehman, Stephen V., and James D. Wickham. 2011. “Pixels, Blocks of Pixels, and Polygons: Choosing a Spatial Unit for Thematic Accuracy Assessment.” Remote Sensing of Environment 115: 3044–3055. doi:10.1016/J.RSE.2011.06.007.
  • Vogeler, Jody C., Robert A. Slesak, Patrick A. Fekety, and Michael J. Falkowski. 2020. “Characterizing over Four Decades of Forest Disturbance in Minnesota, USA.” Forests 11: 362. doi:10.3390/f11030362.

Reviewer 2 Report

This paper developed and applied an approach for attributing agents of forest disturbance across large areas of tropical dry forests, based on Landsat image time-series. This study is interesting and the results may improve the understanding of status and threats of forest degradation. However, there are some concerns that the authors should address before it can be considered for publication.

(1) In order to further highlight the innovation of this article, it is better to compare the results of this study with some related studies.

(2) More mechanism explanations should be added to further explain possible reasons for the different impacts of interference factors on forests in different regions of the study area.

(3) More detailed contents should be added to explain "some of the largest patches in our disturbance map were due to fires 447 (Figure 4)". In addition, the authors should better mark the positions of sub images on the left and right sides of Figure 4.

(4) The discussion about the uncertainty of remote sensing data including forest disturbance map should be added to clarify the limitation of current study. For example, there may be some uncertainties of the results due to the inaccuracy of land use data (e.g. Zhang et al., 2007; Shen et al., 2020; Sevcikova et al., 2021).

(5) This article lacks the conclusion part. I suggest the authors add the separate conclusion part to highlight the key findings of this study.

References:

Zhang et al. The combined use of remote sensing and social sensing data in fine-grained urban land use mapping: A case study in Beijing, China. Remote Sensing, 2017, 9: 865.

Shen et al. Marshland loss warms local land surface temperature in China. Geophysical Research Letters, 2020, 47: e2020GL087648.

Sevcikova H, Nichols B. Land use uncertainty in transportation forecast. Journal of Transport and Land Use, 2021, 14: 805-820.

Round 2

Reviewer 1 Report

In this revision, the authors have made substantial corrections to the manuscript based on my previous comments. I only have several minor comments below.

L104: The authors might use the past tense for “employ”.

L208: It seems that the citation for the harmonization of Landsat ETM+ and OLI SR values is not correct. Perhaps Roy et al. (2016) is suitable here.

L295: Although my recommendation is the stratified random sampling without two-stage sampling, the two-stage sampling procedure is valid as far as the authors adequately consider the inclusion probability of a list of all patches. Although the authors used a minimum distance for reference sample collection, this is not usually required. I do not require the reference collection again, but please see the relevant publications (e.g., Stehman and Foody 2019 section 2.4) for future studies.

L306: Does this mean that reference samples can be labeled as “undisturbed”? If so, how were such samples used for accuracy assessment? Please clarify.

L583-588: Please check the revised sentences. “not fully captured…” (L584)? “we therefore do not…” (L587)?

Reference:

Roy, D. P., V. Kovalskyy, H. K. Zhang, E. F. Vermote, L. Yan, S. S. Kumar, and A. Egorov. 2016. “Characterization of Landsat-7 to Landsat-8 Reflective Wavelength and Normalized Difference Vegetation Index Continuity.” Remote Sensing of Environment 185: 57–70. doi:10.1016/j.rse.2015.12.024.

Stehman, Stephen V., and Giles M. Foody. 2019. “Key Issues in Rigorous Accuracy Assessment of Land Cover Products.” Remote Sensing of Environment 231: 111199. doi:10.1016/J.RSE.2019.05.018.
